# Toto: Time Series Optimized Transformer for Observability

## Abstract

We introduce the Time Series Optimized Transformer for Observability (Toto), a foundation model designed for time series forecasting with a focus on observability metrics. Toto features a novel proportional factorized attention mechanism and a Student-T mixture model head, enabling it to efficiently handle high-dimensional, sparse, and non-stationary data. Trained on one trillion time series data points, including 75% proprietary observability data, Toto demonstrates state-of-the-art zero-shot performance on standard benchmarks such as electricity and weather forecasting. Furthermore, it significantly outperforms existing models in observability-specific tasks, making it an ideal solution for real-time system monitoring and anomaly detection. Toto's architectural innovations make it a versatile tool for both general-purpose forecasting and domain-specific applications, setting a new benchmark for scalability and accuracy in time series analysis.

## 1 Introduction

We present Toto, a time series forecasting foundation model specifically designed to handle the complexities of observability data. It leverages a novel transformer-based architecture to deliver state-of-the-art accuracy and performance. Toto is trained on a massive dataset of diverse time series data, enabling it to excel in zero-shot predictions. Our model is tailored to allow compute and memory-efficient scalability to very large data volumes, thereby providing robust solutions for high-frequency and high-dimensional data commonly encountered in observability metrics.

We detail the following key contributions:

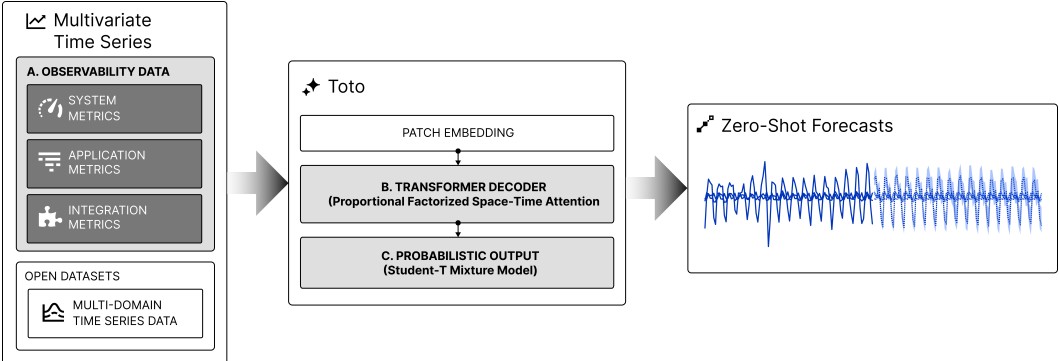

Figure 1: Toto is a novel foundation model for multivariate time series forecasting that achieves state-of-the-art zero-shot accuracy. Key contributions include: **A.** A large-scale pretraining dataset of observability data combined with open time-series datasets; **B.** A novel attention mechanism that enables computationally efficient modeling of both time and space interactions; and **C.** A novel probabilistic prediction head that allows Toto to approximate a wide range of output distributions.

- **Proportional factorized space-time attention:** We introduce an advanced attention mechanism that allows for efficient grouping of multivariate time series features, reducing computational overhead while maintaining high accuracy.

- **Student-T mixture model head:** This novel use of a probabilistic model that robustly generalizes Gaussian mixture models enables Toto to more accurately capture the complex dynamics of time series data and provides superior performance over traditional approaches.

- **Domain-specific training data:** In addition to general multi-domain time series data, Toto is specifically trained on a large-scale dataset of observability metrics, encompassing unique characteristics not present in open-source datasets. This targeted training ensures enhanced performance in observability metric forecasting.

## 1.1 OBSERVABILITY DATA

Observability data encompasses a comprehensive array of metrics collected to monitor and optimize the performance and reliability of modern infrastructure and applications (Li et al., 2020). These metrics are essential for providing insights into the health and performance of systems and include:

- **Infrastructure metrics:** Data related to hardware and system performance, such as memory usage, CPU load, disk I/O, and network throughput.

- **Application performance indicators:** Metrics that capture the performance and behavior of applications, including hit counts, error rates, and latency.

Observability data is typically gathered from a variety of sources, including on-premise systems, cloud services, and third-party tools. The integration of these diverse data sources enables a holistic view of system performance, but also introduces several challenges for time series forecasting:

1. **High temporal resolution:** Observability data often requires high-resolution timestamps, capturing data at intervals of seconds or minutes to detect rapid changes and anomalies.

2. **Sparsity and zero-inflation:** Many observability metrics are sparse, characterized by numerous zero values due to the monitoring of infrequent events, such as system errors or rare performance issues.

3. **Extreme dynamic range and skewed distributions:** Metrics can exhibit wide dynamic ranges and heavy-tailed distributions, especially in latency measurements where occasional extreme values occur.

4. **Dynamic and non-stationary nature:** The monitored systems are dynamic, undergoing frequent changes due to software updates, infrastructure scaling, feature toggles, and varying user behaviors, all of which contribute to non-stationary data patterns.

5. **High-cardinality multivariate data:** Observability data often involves high-dimensional metrics, segmented by various attributes like service type, region, or instance. This results in a large number of time series, each with potentially limited historical data.

6. **Historical anomalies:** Historical data can contain anomalies and outliers resulting from past performance issues or incidents, complicating the forecasting process.

Effectively forecasting observability data requires advanced time series models that can manage these complexities. Traditional forecasting methods often fall short due to their inability to scale and adapt to the dynamic, high-dimensional nature of observability data. Therefore, there is a need for innovative models that can capture intricate patterns and dependencies, ultimately enhancing the ability to proactively detect and mitigate performance issues in real-time systems.

## 1.2 TRADITIONAL MODELS

Historically, time series forecasting has relied on classical models such as ARIMA, exponential smoothing, and basic machine learning techniques (Hyndman & Athanasopoulos, 2021). While foundational, these models necessitate individual training for each metric, presenting several limitations (Fildes et al., 1998). The need to develop and maintain separate models for each metric

impedes scalability, especially given the extensive range of metrics in observability data. Moreover, these models often fail to generalize across different types of metrics, leading to suboptimal performance on diverse datasets (Stevenson, 2007; Christodoulos et al., 2010).

## 1.3 RECENT WORK

Neural models, particularly those based on transformer architectures, have shown promise for improving the accuracy of time series forecasts. These models have demonstrated state-of-the-art performance on benchmark datasets (Nie et al., 2023), frequently surpassing traditional models in both accuracy and robustness. Their capacity to process high-dimensional data efficiently (Lin et al., 2021) makes them ideal for applications involving numerous time series metrics with varying characteristics, such as observability. However, in the full-shot setting, continuous retraining and tuning to adapt to evolving data patterns create a significant operational burden for observability use cases. This scaling limitation has hindered the adoption of deep learning–based methods for time series analysis, even as they show promise in terms of accuracy (Salinas et al., 2020).

Even more recently, a number of time series "foundation models" have been released (Das et al., 2024; Ansari et al., 2024; Woo et al., 2024; Garza & Mergenthaler-Canseco, 2023; Rasul et al., 2023; Gruver et al., 2023). By pre-training on extensive, multi-domain datasets, these large models achieve impressive zero-shot prediction capabilities, significantly reducing the need for constant retraining.

## 1.4 ATTENTION MECHANISMS

To address the unique challenges of time series data, and particularly to adapt transformer architectures for multivariate time-series forecasting, several works have implemented modifications to the attention mechanism. These strategies have included:

- Concatenating variates along the time dimension and computing full self-attention between every space/time location, as in the "any-variate attention" used by Woo et al. (2024). This can capture every possible space and time interaction, but it is costly in terms of computation and memory usage.
- Assuming channel independence, and computing attention only in the time dimension as in Nie et al. (2023). This is efficient, but throws away all information about space-wise interactions.
- Computing attention only in the space dimension, and using a feed-forward network in the time dimension (Ilbert et al., 2024; Liu et al., 2024).
- Computing "factorized attention," where each transformer block contains a separate space and time attention computation (Zhang & Yan, 2023; Rao et al., 2021; Arnab et al., 2021). This allows both space and time mixing, and is more efficient than full cross-attention. However, it doubles the effective depth of the network.

In Section 2.4, we propose a novel approach that allows for both space and time interactions, while reducing the computational cost and improving overall scalability.

## 1.5 PROBABILISTIC OUTPUTS

Practitioners who rely on time series forecasting typically prefer probabilistic predictions. A common practice in neural time series models is to use an output layer where the model regresses the parameters of a probability distribution. This allows for prediction intervals to be computed using Monte Carlo sampling (Salinas et al., 2020).

Common choices for an output layer are Normal (Salinas et al., 2020) and Student-T (Das et al., 2023; Rasul et al., 2023), which can improve robustness to outliers. Moirai (Woo et al., 2024) allows for more flexible residual distributions by proposing a novel mixture model incorporating a weighted combination of Gaussian, Student-T, Log-Normal, and Negative-Binomial outputs.

However, real-world time series can often have complex distributions that are challenging to fit, with outliers, heavy tails, extreme skew, and multimodality. In order to accommodate these scenarios, we

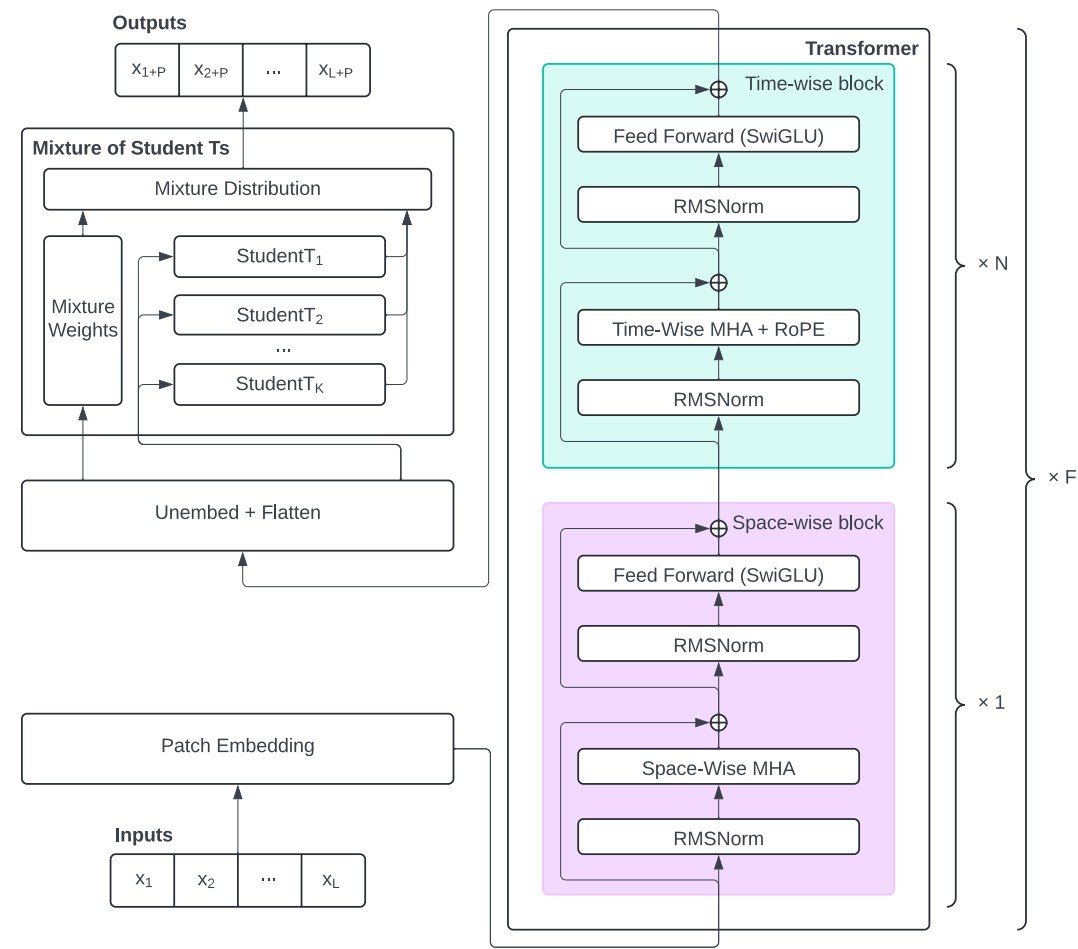

Figure 2: Toto architecture. Input time series of $L$ steps (univariate example used for simplicity here) are first embedded using the patch embedding layer which produces. They then pass through the transformer stack, which contains $F$ identical segments. Each segment of the transformer consists of one space-wise transformer block followed by $N$ time-wise blocks. The flattened transformer outputs are projected to form the parameters of the Student-T mixture model (SMM) head. The final outputs are the forecasts for the input series, shifted $P$ steps (the patch width) into the future.

introduce an even more flexible output likelihood in Section 2.5 based on a Student-T mixture model (Peel & McLachlan, 2000).

## 2 MODEL ARCHITECTURE

### 2.1 TRANSFORMER DESIGN

We build upon the ideas discussed above to define a novel architecture that efficiently models multivariate time series data.

Transformer models for time series forecasting have variously used encoder-decoder (Zhou et al., 2020; Wu et al., 2021; Ansari et al., 2024), encoder-only (Nie et al., 2023; Woo et al., 2024; Liu et al., 2024), and decoder-only architectures (Rasul et al., 2023; Das et al., 2024). For Toto, we employ a decoder-only architecture (Fig. 2). Decoder architectures have been shown to scale well (Radford & Narasimhan, 2018; Radford et al., 2019), and allow for arbitrary prediction horizons. The causal next-patch prediction task also simplifies the pre-training process.

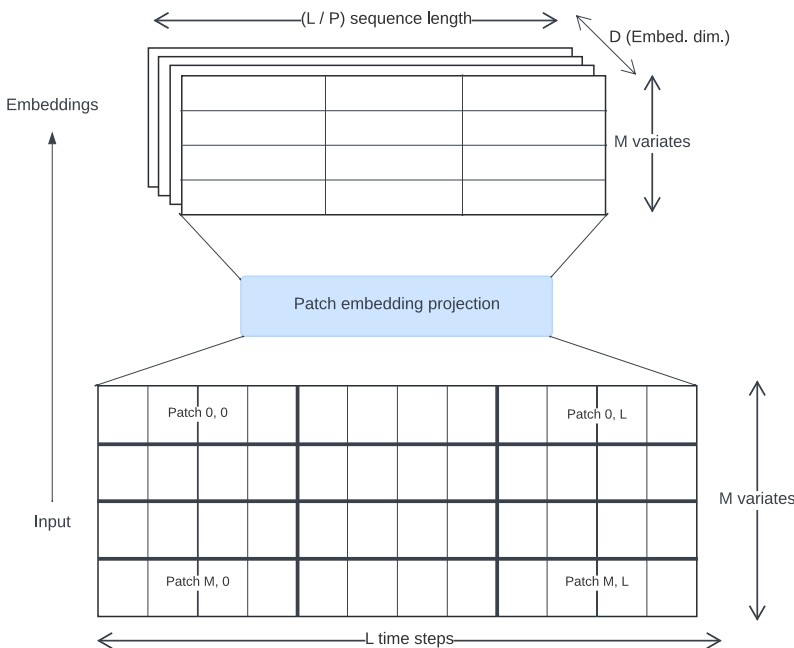

Figure 3: The patch embedding takes as input a multivariate time series of $M$ variates by $L$ time steps. It divides each variate along the time dimension into patches of size $P$ and projects these linearly into an embedding space of latent dimension $D$. This results in an output of size $M \times \frac{L}{P} \times D$ which is fed to the transformer decoder.

We utilize techniques demonstrated to yield performance and efficiency improvements in contemporary transformer literature, including pre-normalization (Xiong et al., 2020), RMSNorm (Zhang & Sennrich, 2019), and SwiGLU feed-forward layers (Shazeer, 2020).

## 2.2 INPUT/OUTPUT SCALING

As in other time series models, we perform instance normalization on input data before passing it through the patch embedding, in order to make the model generalize better to inputs of different scales (Kim et al., 2022). We scale the inputs to have zero mean and unit standard deviation. The output predictions are then rescaled back to the original units.

## 2.3 INPUT EMBEDDING

Time series transformers in the literature have used various approaches for creating input embeddings. We use non-overlapping patch embeddings (Cordonnier et al., 2020; Dosovitskiy et al., 2021; Nie et al., 2023) (Fig. 3) of size $P = 32$, to project input time-series containing $L = 4096$ points to embeddings of size $128 \times D$ per variate, where $D = 512$ is the embedding dimension.

## 2.4 ATTENTION MECHANISM

Observability metrics are often high-cardinality, multivariate time series. Therefore, we designed our model to natively handle multivariate forecasting by analyzing relationships both in the time dimension ("time-wise" interactions) and in the channel dimension ("space-wise" interactions).

In order to model both space and time-wise interactions, we adapt the traditional multi-head attention architecture (Vaswani et al., 2017) from one to two dimensions. We follow the intuition that for many time series, the time relationships are more important or predictive than the space relationships. As evidence, we observe that even models that completely ignore space-wise relationships (such as PatchTST (Nie et al., 2023) and TimesFM (Das et al., 2024)) can still achieve competitive

performance on multivariate datasets. However, other studies (e.g. Woo et al. (2024)) have shown clear benefit to including space-wise attention in ablation studies.

We therefore propose a novel variant of factorized attention, which we call "Proportional Factorized Space-Time Attention." We use a mixture of alternating space-wise and time-wise attention blocks. As a configurable hyperparameter, we can change the ratio of time-wise to space-wise blocks, thus allowing us to devote more or less compute budget to each type of attention. For our base model, we selected a configuration with one space-wise attention block for every two time-wise blocks. This method allows for reduced computational complexity when compared to a traditional attention scheme (see Section A.1).

## 2.5 Probabilistic prediction

In order to produce probabilistic forecasts across the wide range of output distributions present in observability data, we employ a method based on Gaussian mixture models (GMMs), which can approximate any density function (Goodfellow et al., 2016). We find that fitting GMMs leads to numerical instability in training, so we utilize a Student-T mixture model (SMM) of $K$ distributions. This model robustly generalizes GMMs (Peel & McLachlan, 2000), and has previously shown promise for modeling heavy-tailed financial time series (Meitz et al., 2018; WONG et al., 2009). The model predicts $K$ Student-T distributions (where $K$ is a hyperparameter) for each time step, as well as a learned weighting. Formally, the SMM is defined by:

$$p(x) = \sum_{k=1}^{K} \pi_k t(x \mid \mu_k, \tau, \nu_k) \tag{1}$$

where $\pi_{k \in K}$ are nonnegative mixing coefficients which sum to 1 for the $k$th Student's t-distribution $t$ with $\nu_k$ degrees of freedom, mean $\mu_k$, and scale $\sigma_k$. $t(x \mid \mu, \sigma, \nu)$ is defined as:

$$t(x \mid \mu, \tau, \nu) = \frac{\Gamma\left(\frac{\nu+d}{2}\right)}{\Gamma\left(\frac{\nu}{2}\right)(\nu\pi)^{d/2}|\tau|^{1/2}} \left(1 + \frac{1}{\nu}(x-\mu)^T \tau^{-1}(x-\mu)\right)^{-\frac{\nu+d}{2}} \tag{2}$$

When we perform inference, we draw samples from the mixture distribution at each timestamp, then feed each sample back into the decoder for the next prediction. This allows us to produce prediction intervals at any quantile, limited only by the number of samples; for more precise tails, we can choose to spend more computation on sampling (Fig. 4).

As a decoder-only model, Toto is pre-trained on the next-patch prediction task. We minimize the negative log-likelihood of the next predicted patch with respect to the distribution output of the model, defined by the objective function:

$$\text{NLL} = -\log\left(\sum_{k=1}^{16} \pi_k t(x \mid \mu_k, \tau, \nu_k)\right) \tag{3}$$

Additionally, we utilize a dual softmax function on output logits for the mixing coefficients (Cheng et al., 2021), which has been demonstrated to improve training stability with highly heterogeneous data.

We train the model using the AdamW optimizer (Loshchilov & Hutter, 2019). The hyperparameters used for Toto are detailed in Table A1, with 103 million total parameters. In Section A.2, we perform an ablation study on the impact of various model components.

## 3 Training data

We trained Toto with a dataset of approximately one trillion time series points. Of these, roughly three-quarters are anonymous observability metrics from an observability platform. The remaining points come from the LOTSA dataset (Woo et al., 2024), a compilation of publicly-available time

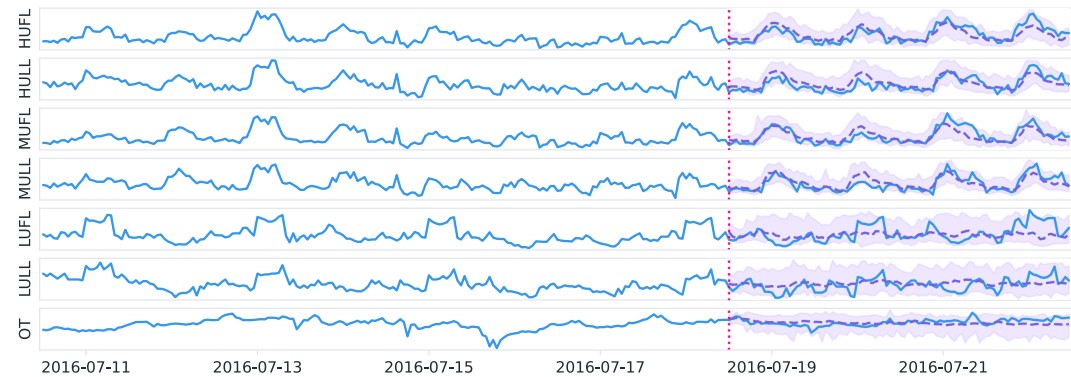

Figure 4: Example of Toto's 96-step zero-shot forecasts on the ETTh1 dataset, showing multivariate probabilistic predictions. Solid lines represent ground truth, dashed lines represent median point forecasts, and shaded regions represent 95% prediction intervals.

series datasets across many different domains. Additionally, we include synthetically generated time series data which we found to improve model performance.

### 3.1 OBSERVABILITY DATASET

An observability platform ingests more than a hundred trillion events per day. However, much of this data is sparse, noisy, or too granular or high in cardinality to be useful in its raw form. To curate a high-quality dataset for efficient model training, we sample queries based on quality and relevance signals from dashboards, monitor alerts, and notebooks. This provides a strong signal that the data resulting from these queries is of critical importance and sufficient quality for observability of real-world applications.

Observability metrics are accessed using a specialized query language supporting filters, group-bys, time aggregation, and various transformations and postprocessing functions. We consider groups returned from the same query to be related variates in a multivariate time series (Fig. A1). After we retrieve the query results, we discard the query strings and group identifiers, keeping only the raw numeric data.

Handling this vast amount of data requires several preprocessing steps to ensure consistency and quality. We describe the details of preprocessing and data augmentation in Section A.3.1.

### 3.2 SYNTHETIC DATA

We use a synthetic data generation process similar to TimesFM (Das et al., 2024) to supplement our training datasets, improving the diversity of the data and helping to teach the model basic structure. The procedure used to generate synthetic data is detailed in Section A.3.2.

## 4 RESULTS

To evaluate predictions, we sequentially divide a time series into context and forecast segments. We input the context segment into Toto and autoregressively generate output patches by sampling from the Student-T mixture model distribution. We forecast a number of steps equal to the nearest multiple of the patch size, then truncate the predictions to the desired length. In order to keep inference time consistent, we vary the number of samples generated based on the cardinality and length of the dataset, with a minimum of 100 samples. We take the median sample at each time step as the final point prediction. This prediction is then compared against the ground-truth forecast segment for evaluation.

We report experimental results for a pre-trained Toto model in Section 4.1 and Section 4.2.

| Dataset | Metric | Zero Shot | | | | | Full Shot | | | | | | | |
|---|---|---|---|---|---|---|---|---|---|---|---|---|---|---|
| | | Toto | Moirai$_{Small}$ | Moirai$_{Base}$ | Moirai$_{Large}$ | TimesFM* | iTransformer | TimesNet | PatchTST | Crossformer | TiDE | DLinear | SCINet | FEDformer |
| ETTh1 | MAE | **0.389** | 0.424 | 0.438 | 0.469 | 0.426 | 0.448 | 0.450 | 0.455 | 0.522 | 0.507 | 0.452 | 0.647 | 0.460 |
| | MSE | **0.363** | 0.400 | 0.434 | 0.510 | - | 0.454 | 0.458 | 0.469 | 0.529 | 0.541 | 0.456 | 0.747 | 0.440 |
| ETTh2 | MAE | **0.261** | 0.379 | 0.382 | 0.376 | 0.410 | 0.407 | 0.497 | 0.407 | 0.684 | 0.550 | 0.515 | 0.723 | 0.449 |
| | MSE | **0.170** | 0.341 | 0.345 | 0.354 | - | 0.383 | 0.414 | 0.387 | 0.942 | 0.611 | 0.559 | 0.954 | 0.437 |
| ETTm1 | MAE | **0.375** | 0.409 | 0.388 | 0.389 | 0.388 | 0.410 | 0.406 | 0.400 | 0.495 | 0.419 | 0.407 | 0.481 | 0.452 |
| | MSE | **0.372** | 0.448 | 0.381 | 0.390 | - | 0.407 | 0.400 | 0.387 | 0.513 | 0.419 | 0.403 | 0.486 | 0.448 |
| ETTm2 | MAE | **0.319** | 0.341 | 0.321 | 0.320 | 0.334 | 0.332 | 0.333 | 0.326 | 0.611 | 0.404 | 0.401 | 0.537 | 0.349 |
| | MSE | 0.272 | 0.300 | **0.272** | 0.276 | - | 0.288 | 0.291 | 0.281 | 0.757 | 0.358 | 0.350 | 0.571 | 0.305 |
| Electricity | MAE | **0.246** | 0.320 | 0.274 | 0.273 | - | 0.270 | 0.295 | 0.304 | 0.334 | 0.344 | 0.300 | 0.365 | 0.327 |
| | MSE | **0.157** | 0.233 | 0.188 | 0.188 | - | 0.178 | 0.193 | 0.216 | 0.244 | 0.252 | 0.212 | 0.268 | 0.214 |
| Weather | MAE | 0.284 | 0.267 | **0.261** | 0.275 | - | 0.278 | 0.287 | 0.281 | 0.315 | 0.320 | 0.317 | 0.363 | 0.360 |
| | MSE | 0.256 | 0.242 | **0.238** | 0.259 | - | 0.258 | 0.259 | 0.259 | 0.259 | 0.271 | 0.265 | 0.292 | 0.309 |
| Mean | MAE | **0.312** | 0.357 | 0.341 | 0.350 | - | 0.357 | 0.378 | 0.362 | 0.493 | 0.424 | 0.399 | 0.519 | 0.400 |
| | MSE | **0.265** | 0.328 | 0.315 | 0.330 | - | 0.328 | 0.336 | 0.333 | 0.541 | 0.409 | 0.374 | 0.533 | 0.359 |

Table 1: Comparison of different models with Toto on the LSF benchmark datasets. Results are averaged across prediction lengths of 96, 192, 336, and 720 steps. For Toto, we use a stride of 512 steps and a historical context window of 512 steps. For other models, we use the results reported in Woo et al. (2024) and Das et al. (2024). Metrics for each prediction length are available in Table A3. *TimesFM only reports values for MAE on ETTh1, ETTh2, ETTm1, and ETTm2.
Key: **Best results**, Second-best results.

## 4.1 LSF BENCHMARKS

To assess general-purpose time series forecasting performance, we use the Long Sequence Forecasting (LSF) benchmark datasets (ETTh1, ETTh2, ETTm1, ETTm2, Electricity, and Weather) (Wu et al., 2021). For Toto, we used a historical context window of 512 time steps and took the median of 200 samples. Following standard practice, we report normalized Mean Absolute Error (MAE) and Mean Squared Error (MSE), fitted on a training split, in order to be able to compare performance across different datasets. We compared Toto's performance with the reported results of other recent zero-shot foundation models (Woo et al., 2024; Das et al., 2024), as well as full-shot time series forecasting models (Liu et al., 2024; Wu et al., 2023; Nie et al., 2023; Zhang & Yan, 2023; Das et al., 2023; Zeng et al., 2023; LIU et al., 2022; Zhou et al., 2022). We evaluate with forecast lengths of 96, 192, 336, and 720 time steps, in sliding windows with stride 512, and average the results. We display these results in Table 1.

Toto demonstrates exceptional performance across a variety of benchmark datasets, excelling in zero-shot scenarios. In the LSF datasets, Toto consistently outperforms other models in terms of MAE and MSE. For example, on the ETTh1 dataset, Toto achieves an MAE of 0.389 and an MSE of 0.363, outperforming all zero-shot models, including the previously reported Moirai series and TimesFM. Macro-averaging across the six LSF datasets, Toto achieves an MAE of 0.312 and MSE of 0.265, again exceeding Moirai's reported zero-shot performance as well as the reported performance of the full-shot models.

While Toto generally excels, there are areas where its performance is closely matched by other models. In full-shot scenarios, models like PatchTST, Crossformer, and FEDformer show competitive results. For example, on the Electricity dataset, while Toto achieves a leading zero-shot MAE of 0.246 and MSE of 0.157, iTransformer and TimesNet also show strong performance, indicating that these models can catch up when additional training data is available.

## 4.2 OBSERVABILITY BENCHMARK

We created a benchmark using anonymous observability data to assess performance across various observability metrics. To ensure a representative and realistic sample, we sampled data based on quality and relevance signals from dashboards, monitor alerts, and notebooks. This benchmark comprises 983,994 data points from 82 distinct multivariate time series, encompassing 1,122 variates.

We analyzed summary statistics of the series in our benchmark to identify characteristics that make observability time series challenging to forecast. The categories and their definitions are as follows:

- **Sparse:** Series with a low density of observations, indicating infrequent recording of data or rare events.

| Metric | Toto | Chronos-T5$_{Tiny}$ | Chronos-T5$_{Mini}$ | Chronos-T5$_{Small}$ | Chronos-T5$_{Base}$ | Chronos-T5$_{Large}$ | Moirai$_{Small}$ | Moirai$_{Base}$ | Moirai$_{Large}$ | TimesFM |
|---|---|---|---|---|---|---|---|---|---|---|
| sMAPE | **0.672** | 0.809 | 0.788 | 0.800 | 0.796 | 0.805 | 0.808 | 0.742 | 0.736 | 1.246 |
| sMdAPE | **0.318** | 0.406 | 0.391 | 0.401 | 0.393 | 0.396 | 0.418 | 0.370 | 0.365 | 0.639 |

Table 2: Performance of Toto and other zero-shot models on the observability benchmark dataset. Key: **Best results**, Second-best results.

- **Extreme right skew:** Series with a distribution heavily skewed to the right, characterized by a few very high values and many lower values.

- **Seasonal:** Series exhibiting regular and recurring patterns, often linked to daily, weekly, or yearly cycles.

- **Flat:** Series with minimal variability, showing little to no change over time.

To assess the prediction of other zero-shot models on the observability Benchmark, we follow sampling procedures delineated in their respective manuscripts. In short, for Chronos models, we generate 20 samples and take the median prediction. For Moirai models, we take the median of 100 samples and set the patch size to "auto". TimesFM only produces point predictions of the mean, so we use those directly. Since TimesFM and Chronos only support univariate forecasting, we process each variate independently. Moirai, on the other hand, like Toto, makes joint predictions for each group of related variates. For Toto, we utilize the same evaluation procedure we used on the LSF benchmarks.

The relative proportion of these cases are displayed in Table A4. The evaluation results (Table 2) demonstrate that Toto outperforms the other models.

Because observability data can have extreme variation in both magnitude and dispersion, we select symmetric mean absolute percentage error (sMAPE) as a scale-invariant performance metric (Armstrong, 1985). We also report symmetric median absolute percentage error (sMdAPE), a robust version of sMAPE (Hyndman & Koehler, 2006) that minimizes the influence of the extreme outliers present in observability data. With the lowest sMAPE of 0.672 and sMdAPE of 0.318, Toto proves to be the most accurate for forecasting observability time series data.

These results suggest that current open datasets may not provide sufficient information to extrapolate to the specific nuances of observability data, highlighting the importance of training on more relevant data as demonstrated by Toto's superior performance.

## 5 CONCLUSIONS

Toto demonstrates state-of-the-art performance across both public and proprietary benchmarks. By leveraging a proportional factorized attention mechanism and a Student-T mixture model, Toto achieves impressive results in both zero-shot and full-shot settings, showcasing its scalability and flexibility in handling complex, high-dimensional data.

Despite its success, there are areas where further refinement is possible. Future work could involve integrating additional input modalities or exploring new attention mechanisms to enhance scalability and accuracy.

With its demonstrated robustness and ability to manage observability data at scale, Toto not only advances time series forecasting but also opens new pathways for real-time system monitoring and infrastructure optimization, setting the stage for further innovations in the field.

## 6 IMPACT STATEMENT

In developing Toto, we followed a structured approach to ensure responsible development, focusing on identifying, assessing, and mitigating potential risks associated with the use of our model. Given that Toto is not intended for mass distribution and specifically generates time series forecasts for observability data, the potential harms are considerably lower compared to more general-purpose models. Our primary focus was ensuring the accuracy, reliability, and security of the forecasts

generated by Toto, which are crucial for maintaining and optimizing infrastructure and application performance.

We carefully analyze the potential for Toto to produce incorrect or misleading forecasts that could impact decision-making processes in critical systems. Additionally, we consider the implications of Toto's performance across diverse datasets, ensuring it can generalize well without introducing significant errors.

## 7 FUTURE DIRECTIONS

Many exciting areas of exploration remain for further study. Some future research questions that particularly intriguing include:

- **Multi-modal inputs:** Incorporate additional input modalities such as query metadata and captions to enhance forecast performance.
- **Autonomous troubleshooting agents:** Creating AI agents for troubleshooting and incident response by integrating modality-specific foundation models like Toto to improve their reasoning and planning capabilities with telemetry data.
- **Conversational interfaces:** Align time series forecasting models with LLMs to develop conversational agents capable of interpreting and reasoning about time series data.
- **Model enhancements and scaling:** Explore ways to improve and scale model performance through optimizations such as new types of input embeddings, attention mechanisms, and examining alternative variate groupings to capture richer interactions.

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

# A APPENDIX

## A.1 MODEL ARCHITECTURE

After the patchwise embedding layer, we have inputs of shape $\mathbf{X} \in \mathbb{R}^{B \times M \times \frac{L}{P} \times D}$, where $B$ is the batch dimension, $M$ is the number of variates per batch item, $\frac{L}{P}$ is time steps divided by patch width, and $D$ is the model embedding dimension.

**Time-wise attention.** We parallelize along the time dimension by reshaping the input tensor:

$$\mathbf{X} \in \mathbb{R}^{B \times M \times \frac{L}{P} \times D} \to \mathbf{X}_{\text{time}} \in \mathbb{R}^{(B \times M) \times \frac{L}{P} \times D}$$

This allows for attention to be calculated independently in parallel per variate, giving a complexity of:

$$\mathcal{O}(M \times (\frac{L}{P})^2 \times D)$$

In the time-wise attention blocks, we use causal masking and rotary positional embeddings (Su et al., 2021) with XPOS (Sun et al., 2022) in order to autoregressively model time-dependent features.

**Space-wise attention.** We similarly parallelize along the time dimension by reshaping the input tensor:

$$\mathbf{X} \in \mathbb{R}^{B \times M \times \frac{L}{P} \times D} \to \mathbf{X}_{\text{space}} \in \mathbb{R}^{(B \times \frac{L}{P}) \times M \times D}$$

We calculate attention in parallel for each time step, with complexity:

$$\mathcal{O}(\frac{L}{P} \times M^2 \times D)$$

In the space-wise blocks, we use full bidirectional attention (without causal masking) in order to preserve permutation invariance of the covariates, with a block-diagonal ID mask to ensure that only related variates attend to each other. This masking allows us to pack multiple independent multivariate time series into the same batch, in order to improve training efficiency and reduce the amount of padding.

**Computational complexity.** Each transformer block in our model contains $N$ timewise attention layers and 1 spacewise layer. The complexity for full self-attention over $N + 1$ layers, where interactions can occur across all variates and sequence positions, would be of complexity:

$$\mathcal{O}\left( (N+1) \times M^2 \times \left(\frac{L}{P}\right)^2 \times D \right) \tag{A1}$$

This reflects the quadratic dependence on both the sequence length $\frac{L}{P}$ and the variate dimension $M$, with linear dependence on the embedding dimension $D$. However, by utilizing factorized attention, we can reduce the computational complexity of the attention calculation to:

$$\mathcal{O}\left( N \times M \times \left(\frac{L}{P}\right)^2 \times D + \frac{L}{P} \times M^2 \times D \right) =$$
$$\mathcal{O}\left( D \times \frac{L}{P} \times M \times \left( N \times \frac{L}{P} + M \right) \right) \tag{A2}$$

We demonstrate that factorized space-wise attention is asymptotically smaller in computational complexity than full self-attention (see Equation A1 and Equation A2). When comparing a model with full self-attention, we can assume $N$ and $D$ are fixed. Therefore:

$$\mathcal{O}\left(M \times \left(\frac{L}{P}\right)^2 + \frac{L}{P} \times M^2\right) < \mathcal{O}\left(M^2 \times \left(\frac{L}{P}\right)^2\right)$$

which reduces to:

$$\mathcal{O}\left(M + \frac{L}{P}\right) < \mathcal{O}\left(M \times \frac{L}{P}\right)$$

Thus, by factorizing attention into time-wise and space-wise components, the computational complexity is reduced, especially for large numbers of variates $M$ or long sequences $\frac{L}{P}$, making it more scalable than full self-attention.

### A.1.1 HYPERPARAMETERS

| Hyperparameter | Value |
|---|---:|
| Embedding Dimension | 512 |
| MLP Dimension | 2048 |
| # Layers | 24 |
| # Heads | 8 |
| # Variates | 32 |
| $(\beta_1, \ \beta_2)$ | (0.9, 0.95) |
| Weight Decay | 0.01 |
| Spacewise Layer Cadence | 3 |
| Patch Size | 32 |
| # Student-T Mixture Model Components | 16 |
| Initial Learning Rate | 0.001 |
| Annealing Schedule | Cosine |
| Batch Size | 192 |
| Warmup Steps | 5000 |
| Total Train Steps | 193000 |

Table A1: Hyperparameters for Toto

### A.2 ABLATIONS

In this ablation study, we compare several versions of the Toto model using Negative Log Likelihood (NLL) loss on the validation set of our observability dataset. In addition to the full Toto model, we train separate variants with:

1. No space-wise attention (Time-wise Attention layers only)

2. No Student-T mixture model (instead, we replace the output with a single Student-T distribution)

3. No observability data (instead, we train only on the full LOTSA dataset with synthetic data)

All models (except the "no observability data" model) were trained on a scaled down dataset with 620B points, with the number of training steps proportionally reduced to 117,000 steps. For each model, we report the NLL at its minimum during training and present the relative performance as a percentage decrease in comparison to the full Toto model. Table A2 presents the performance of each model variant, showing the percentage increase in NLL relative to the full Toto model (lower percentages indicate worse performance).

| Model | NLL (% Increase Relative to Toto) |
|---|---|
| Toto | (baseline) 0% |
| No Space-wise Attention | 4.37% |
| Single Student-T | 11.48% |
| No Observability Data | 14.21% |

Table A2: Percentage increase in NLL relative to the full Toto model.

We observe that the full Toto model achieves the lowest NLL at its best validation point, serving as the baseline. The "No Space-wise Attention" variant shows a 4.37% increase in NLL, while the "Single Student-T" and "No observability Data" variants show larger decreases in performance, with NLL increases of 11.48% and 14.21%, respectively. These results indicate that space-wise attention, the Student-T mixture model, and the inclusion of observability -specific data are essential for optimal model performance. The percentage differences highlight the impact of these components on the model's ability to effectively model the underlying data distribution.

### A.3 TRAINING DATA PREPROCESSING

#### A.3.1 OBSERVABILITY DATASET

Initially, we apply padding and masking techniques to align the series lengths, making them divisible by the patch stride. This involves adding necessary left-padding to both the time series data and the ID mask, ensuring compatibility with the model's requirements.

Various data augmentations are employed to enhance the dataset's robustness. We introduce random time offsets to prevent memorization caused by having series always align the same way with the patch grid. After concatenating the observability and LOTSA datasets for training, we also implement a variate shuffling strategy to maintain diversity and representation. Specifically, we randomly combine variates from either observability , LOTSA, and/or synthetic data with a probability of 10%, thus creating new, diverse combinations of data points. To sample the indices when mixing in this fashion, we employ a normal distribution with a standard deviation of 1000, favoring data points that were closer together in the original datasets. This Gaussian sampling ensures that, while there is a preference for adjacent data points, significant randomness is introduced to enhance the diversity of the training data. This approach improves the model's ability to generalize across different types of data effectively.

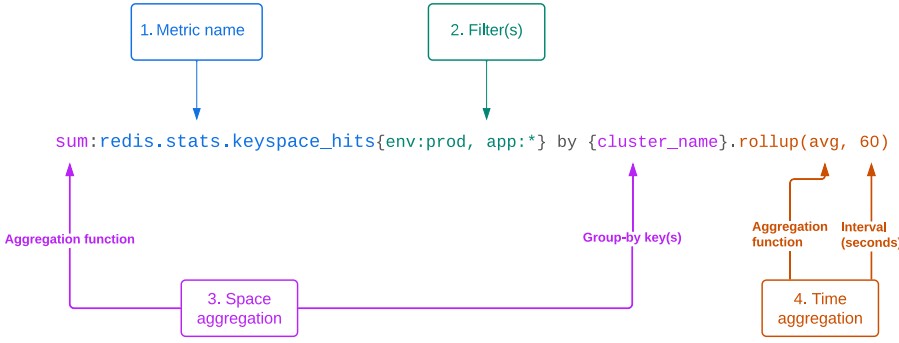

Figure A1: Example metric query in the observability platform. The metric name (1) determines which metric is being queried. The filter clause (2) limits which contexts are queried, in this case restricting the query to the prod environment. The space aggregation (3) indicates that the average metric value should be returned for each unique combination of the group-by keys. The time aggregation (4) indicates that metric values should be aggregated to the average for each 60-second interval. The query results will be a multivariate time series with 1-minute time steps, and with separate individual variates for each unique service, datacenter tuple.

### A.3.2 SYNTHETIC DATA

We simulate time series data through the composition of components such as piecewise linear trends, ARMA processes, sinusoidal seasonal patterns, and various residual distributions. We randomly combine five of these processes per variate, introducing patterns not always present in our real-world datasets. The generation process involves creating base series with random transformations, clipping extreme values, and rescaling to a specified range. By making synthetic data approximately 5% of our training dataset, we ensure a wide range of time-series behaviors are captured. This diversity exposes our models to various scenarios during training, improving their ability to generalize and effectively handle real-world data.

## A.4 RESULTS

### A.4.1 LSF BENCHMARKS

| Dataset | Prediction Length | Metric | Zero Shot | | | | | Full Shot | | | | | | | |
|---|---|---|---|---|---|---|---|---|---|---|---|---|---|---|---|
| | | | Toto | Moirai$_{Small}$ | Moirai$_{Base}$ | Moirai$_{Large}$ | TimesFM | iTransformer | TimesNet | PatchTST | Crossformer | TiDE | DLinear | SCINet | FEDformer |
| ETTh1 | 96 | MAE | **0.366** | 0.402 | 0.402 | 0.398 | 0.398 | 0.405 | 0.402 | 0.419 | 0.448 | 0.464 | 0.400 | 0.599 | 0.419 |
| | | MSE | **0.307** | 0.375 | 0.384 | 0.380 | - | 0.386 | 0.384 | 0.414 | 0.423 | 0.479 | 0.386 | 0.654 | 0.376 |
| | 192 | MAE | **0.368** | 0.419 | 0.429 | 0.434 | 0.424 | 0.436 | 0.429 | 0.445 | 0.474 | 0.492 | 0.432 | 0.631 | 0.448 |
| | | MSE | **0.329** | 0.399 | 0.425 | 0.440 | - | 0.441 | 0.436 | 0.460 | 0.471 | 0.525 | 0.437 | 0.719 | 0.420 |
| | 336 | MAE | **0.399** | 0.429 | 0.450 | 0.474 | 0.436 | 0.458 | 0.469 | 0.466 | 0.546 | 0.515 | 0.459 | 0.659 | 0.465 |
| | | MSE | **0.396** | 0.412 | 0.456 | 0.514 | - | 0.487 | 0.491 | 0.501 | 0.570 | 0.565 | 0.481 | 0.778 | 0.459 |
| | 720 | MAE | **0.424** | 0.444 | 0.473 | 0.568 | 0.445 | 0.491 | 0.500 | 0.488 | 0.621 | 0.558 | 0.516 | 0.699 | 0.507 |
| | | MSE | 0.419 | **0.413** | 0.470 | 0.705 | - | 0.503 | 0.521 | 0.500 | 0.653 | 0.594 | 0.519 | 0.836 | 0.506 |
| ETTh2 | 96 | MAE | **0.197** | 0.334 | 0.327 | 0.325 | 0.356 | 0.349 | 0.374 | 0.348 | 0.584 | 0.400 | 0.387 | 0.621 | 0.397 |
| | | MSE | **0.093** | 0.281 | 0.277 | 0.287 | - | 0.297 | 0.340 | 0.302 | 0.745 | 0.400 | 0.333 | 0.707 | 0.358 |
| | 192 | MAE | **0.231** | 0.373 | 0.374 | 0.367 | 0.400 | 0.400 | 0.414 | 0.400 | 0.656 | 0.509 | 0.476 | 0.689 | 0.439 |
| | | MSE | **0.135** | 0.340 | 0.340 | 0.347 | - | 0.380 | 0.402 | 0.388 | 0.877 | 0.528 | 0.477 | 0.860 | 0.429 |
| | 336 | MAE | **0.260** | 0.393 | 0.401 | 0.393 | 0.428 | 0.432 | 0.541 | 0.433 | 0.731 | 0.541 | 0.541 | 0.744 | 0.487 |
| | | MSE | **0.160** | 0.362 | 0.371 | 0.377 | - | 0.428 | 0.452 | 0.426 | 1.043 | 0.643 | 0.594 | 1.000 | 0.496 |
| | 720 | MAE | **0.355** | 0.416 | 0.426 | 0.421 | 0.457 | 0.445 | 0.657 | 0.446 | 0.763 | 0.679 | 0.657 | 0.838 | 0.474 |
| | | MSE | **0.294** | 0.380 | 0.394 | 0.404 | - | 0.427 | 0.462 | 0.431 | 1.104 | 0.874 | 0.831 | 1.249 | 0.463 |
| ETTm1 | 96 | MAE | **0.328** | 0.383 | 0.360 | 0.363 | 0.345 | 0.368 | 0.375 | 0.367 | 0.426 | 0.387 | 0.372 | 0.438 | 0.419 |
| | | MSE | **0.306** | 0.404 | 0.335 | 0.353 | - | 0.334 | 0.338 | 0.329 | 0.404 | 0.364 | 0.345 | 0.418 | 0.379 |
| | 192 | MAE | **0.353** | 0.402 | 0.379 | 0.380 | 0.374 | 0.391 | 0.387 | 0.385 | 0.451 | 0.404 | 0.389 | 0.450 | 0.441 |
| | | MSE | **0.328** | 0.435 | 0.366 | 0.376 | - | 0.377 | 0.374 | 0.367 | 0.450 | 0.398 | 0.380 | 0.439 | 0.426 |
| | 336 | MAE | **0.389** | 0.416 | 0.394 | 0.395 | 0.397 | 0.420 | 0.411 | 0.410 | 0.515 | 0.425 | 0.413 | 0.485 | 0.459 |
| | | MSE | **0.390** | 0.462 | 0.391 | 0.399 | - | 0.426 | 0.410 | 0.399 | 0.532 | 0.428 | 0.413 | 0.490 | 0.445 |
| | 720 | MAE | 0.429 | 0.437 | 0.419 | **0.417** | 0.436 | 0.459 | 0.450 | 0.454 | 0.589 | 0.453 | 0.453 | 0.550 | 0.490 |
| | | MSE | 0.463 | 0.490 | 0.434 | **0.432** | - | 0.491 | 0.478 | 0.454 | 0.666 | 0.487 | 0.474 | 0.595 | 0.543 |
| ETTm2 | 96 | MAE | 0.270 | 0.282 | 0.269 | 0.260 | 0.263 | 0.264 | 0.267 | **0.259** | 0.366 | 0.305 | 0.292 | 0.377 | 0.287 |
| | | MSE | 0.200 | 0.205 | 0.195 | 0.189 | - | 0.180 | 0.187 | **0.175** | 0.287 | 0.207 | 0.193 | 0.286 | 0.203 |
| | 192 | MAE | 0.315 | 0.318 | 0.303 | **0.300** | 0.309 | 0.309 | 0.309 | 0.302 | 0.492 | 0.364 | 0.362 | 0.445 | 0.328 |
| | | MSE | 0.269 | 0.261 | 0.247 | 0.247 | - | 0.250 | 0.249 | **0.241** | 0.414 | 0.290 | 0.284 | 0.399 | 0.269 |
| | 336 | MAE | 0.319 | 0.355 | 0.333 | 0.334 | 0.349 | 0.348 | 0.351 | 0.343 | 0.542 | 0.427 | 0.427 | 0.591 | 0.366 |
| | | MSE | **0.264** | 0.319 | 0.291 | 0.295 | - | 0.311 | 0.321 | 0.305 | 0.597 | 0.377 | 0.369 | 0.637 | 0.325 |
| | 720 | MAE | **0.374** | 0.410 | 0.377 | 0.386 | 0.415 | 0.407 | 0.403 | 0.400 | 1.042 | 0.524 | 0.522 | 0.735 | 0.415 |
| | | MSE | 0.354 | 0.415 | 0.355 | 0.372 | - | 0.412 | 0.408 | 0.402 | 1.730 | 0.558 | 0.554 | 0.960 | 0.421 |
| Electricity | 96 | MAE | **0.212** | 0.299 | 0.248 | 0.242 | - | 0.240 | 0.272 | 0.285 | 0.314 | 0.329 | 0.282 | 0.345 | 0.308 |
| | | MSE | **0.124** | 0.205 | 0.158 | 0.152 | - | 0.148 | 0.168 | 0.195 | 0.219 | 0.237 | 0.197 | 0.247 | 0.193 |
| | 192 | MAE | **0.232** | 0.310 | 0.263 | 0.259 | - | 0.253 | 0.289 | 0.289 | 0.322 | 0.330 | 0.285 | 0.355 | 0.315 |
| | | MSE | **0.138** | 0.220 | 0.174 | 0.171 | - | 0.162 | 0.184 | 0.199 | 0.231 | 0.236 | 0.196 | 0.257 | 0.201 |
| | 336 | MAE | **0.249** | 0.323 | 0.278 | 0.278 | - | 0.269 | 0.300 | 0.305 | 0.337 | 0.344 | 0.301 | 0.369 | 0.329 |
| | | MSE | **0.155** | 0.236 | 0.191 | 0.192 | - | 0.178 | 0.198 | 0.215 | 0.246 | 0.249 | 0.209 | 0.269 | 0.214 |
| | 720 | MAE | **0.291** | 0.347 | 0.307 | 0.313 | - | 0.317 | 0.320 | 0.337 | 0.363 | 0.373 | 0.333 | 0.390 | 0.355 |
| | | MSE | **0.211** | 0.270 | 0.229 | 0.236 | - | 0.225 | 0.220 | 0.256 | 0.280 | 0.284 | 0.245 | 0.299 | 0.246 |
| Weather | 96 | MAE | 0.223 | 0.212 | **0.203** | 0.208 | - | 0.214 | 0.220 | 0.218 | 0.230 | 0.261 | 0.255 | 0.306 | 0.296 |
| | | MSE | 0.180 | 0.173 | 0.167 | 0.177 | - | 0.174 | 0.172 | 0.177 | **0.158** | 0.202 | 0.196 | 0.221 | 0.217 |
| | 192 | MAE | 0.267 | 0.250 | **0.241** | 0.249 | - | 0.254 | 0.261 | 0.259 | 0.277 | 0.298 | 0.296 | 0.340 | 0.336 |
| | | MSE | 0.235 | 0.216 | 0.209 | 0.219 | - | 0.221 | 0.219 | 0.225 | **0.206** | 0.242 | 0.237 | 0.261 | 0.276 |
| | 336 | MAE | 0.291 | 0.282 | **0.276** | 0.292 | - | 0.296 | 0.306 | 0.297 | 0.335 | 0.335 | 0.335 | 0.378 | 0.380 |
| | | MSE | 0.252 | 0.260 | 0.256 | 0.277 | - | 0.278 | 0.280 | 0.278 | 0.272 | 0.287 | 0.283 | 0.309 | 0.339 |
| | 720 | MAE | 0.356 | **0.322** | 0.323 | 0.350 | - | 0.349 | 0.359 | 0.348 | 0.418 | 0.386 | 0.381 | 0.427 | 0.428 |
| | | MSE | 0.356 | 0.320 | 0.321 | 0.365 | - | 0.358 | 0.365 | 0.354 | 0.398 | 0.351 | 0.345 | 0.377 | 0.403 |

Table A3: Performance metrics for various models. Key: **Best results**, Second-best results.

### A.4.2 OBSERVABILITY BENCHMARK

| Case | % |
|---|---|
| Sparse | 12.20 |
| Extreme Right Skew | 17.07 |
| Seasonal | 80.49 |
| Flat | 1.22 |

Table A4: Breakdown of observability dataset based on case, computed based on the average characteristics of variates in each multivariate series. Note that these do not add to 100% because time series may fall into multiple categories.

