# OpenReview forum: "Toto: Time Series Optimized Transformer for Observability"
_ICLR.cc/2025/Conference — ICLR 2025 Conference Withdrawn Submission_

### Official Review · Reviewer_NzBv · 2024-10-21

**Soundness:** 2
**Presentation:** 3
**Contribution:** 1
**Rating:** 3
**Confidence:** 5

**Summary:**

The paper proposes a pretrained time series forecasting model with a focus on observability time series data. Toto is a decoder-only patch-based model trained on large amounts of time series data to predict the next patch of time series. Due to its focus on observability, the training corpus includes a large proportion of proprietary observability time series. Toto uses a mixture of Student's-t distribution head to enable forecasting of heavy-tailed and non-stationary data. It includes a factorized "space-time" attention mechanism allowing it to consume multivariate time series and generate forecasts jointly. Experimental results have been reported on some datasets from the long-term forecasting benchmark and a proprietary observability benchmark.

**Strengths:**

The paper is well written and provides an insightful discussion on the specific case of observability time series data. Some design choices such as the factorized attention mechanism and mixture of student's-t head have been reasonably justified.

**Weaknesses:**

- The paper proposes another pretrained model for time series such as Chronos, TimesFM and Moirai. When compared with these existing models, the technical contribution of Toto is marginal. The mixture of distributions idea is not new and has been studied in forecasting literature before, most recently even in the context of pretrained models (Moirai). Furthermore, the alternating spatial and temporal attention blocks idea is incremental.

- In the absence of strong methodological contributions, the empirical investigation needs to be strong to meet the acceptance bar. Unfortunately, the experiments reported in this paper are sub par. Concretely, they have the following limitations.
    - The breadth of experiments is severely limited for a purported "foundation model". In the benchmark on publicly available datasets (Table 1), only 6 datasets have been studied, 5 of which belong to the same domain with 4 being essentially the same dataset (ETTh1, ETTh2, ETTm1, ETTm2). This benchmark is limited even for a regular time series paper. For a pretrained model, this is not sufficient to draw any conclusions. Please check the Chronos and/or TimesFM papers for a better evaluation benchmark.
    - Even on these 6 datasets, it appears that the numbers reported for Toto are not from the same setting as other methods. The numbers for baselines such as Moirai and TimesFM have been reproduced from their papers but the rolling evaluation for Toto appears to have been conducted with a stride of 512 which is not the same setting as in the baselines. Due to the fewer slices being evaluated, the numbers could be inflated. Please clarify if I have misunderstood the setting. See questions for details.
    - The second benchmark for observability time series appears to have been conducted on a proprietary dataset which makes it difficult to verify for the community. Will the dataset or the pretrained model weights be released? In either case, code/model could have been made available to the reviewers. Regardless of reproducibility concerns, the percentage metrics reported are often discouraged in the literature (see [1] for details). Furthermore, it is unclear what the key claim of this experiment is. Since Toto has been trained on large amounts of data from the same observability domain (75% of 1T time series), it's not surprising that it would perform better than other "zero-shot" models.
    - Beyond reporting the numbers, the paper does not conduct any serious investigation of the models or the results.

- Minor: It's not clear to me what a "foundation model" for time series forecasting in observability really means. With such domain-specific qualifier, the word "foundation" loses its meaning and it would be better replaced with something like "pretrained".

[1] Hyndman and Koehler (2006). Another look at measures of forecast accuracy.

**Questions:**

- What is meant by "... in sliding windows with stride 512 ..." in line 406? Can you share how many windows you end up with for different prediction lengths? (Let's take ETTh1 as an example)
- I am willing to increase my score if the authors expand their evaluation benchmark. A reasonably simple way to do this would be to report results on the benchmark presented in the Chronos paper. As per my knowledge, the code for this benchmark is available on Github.

---

### Official Review · Reviewer_9pPH · 2024-10-31

**Soundness:** 2
**Presentation:** 3
**Contribution:** 2
**Rating:** 3
**Confidence:** 3

**Summary:**

The this paper introduces a time series foundation model for observability data (called Toto). The model utilizes a factorized-space time transformer variant to handle multivariate time series and a mixture student-t head for probabilistic forecasting. The model is trained on a large dataset of proprietary observability data in addition to open source, and synthetic data. The authors evaluate their method on a long sequence forecasting benchmark and on the observability dataset.

**Strengths:**

The main strength of the paper of the paper is that it introduces a foundation model specifically for the observability domain, which is an important practical problem. Using a foundation model that can be rolled out broadly on an entire system without local model fitting or retraining is a sound approach to address this problem [**Originality and Significance**]. The paper is clearly written and easy to follow [**Clarity**].

**Weaknesses:**

The two main weaknesses of this manuscript are that the architectural contributions are incremental and the validation on the long sequence forecasting benchmark requires either revision or clarification.

**Incremental contribution**: I would argue that the main contribution of this paper is the observability dataset. A high quality dataset for this domain would be helpful to further develop foundation models for observability specifically and for time series foundation models more generally. However, since the data in this work is proprietary this will not benefit the ICLR or larger science community. The other contributions (proportional factorized space-time attention) and the mixture student-t probabilistic output distribution are incremental. The proposed transformer architecture is a minor modification from Arnab et al. ICCV 2021 (https://arxiv.org/abs/2103.15691) with a configurable parameter to adjust the ratio of space and time attention layers. The mixture student-t distribution is a minor modification of the mixture distribution output in Woo et al. ICML 2024 (https://arxiv.org/abs/2402.02592). GluonTS (Alexandrov et al., JMLR 2020) also provides an implementation of a generic mixture distribution that can be instantiated as a student-t mixture.

Thus, I would regard the contributions as too incremental to warrant publication in ICLR.

**Evaluation on Long Sequence Forecasting (LSF) Dataset**: Another concern is on the evaluation on the LSF dataset. The setup in this manuscript uses a rolling evaluation with a stride 512, which is inconsistent with the results taken from earlier work that is included in this evaluation (Woo et al. ICML 2024). Specifically, the original setup of this dataset uses a stride of 1 (Zhou et al., AAAI 2021: https://arxiv.org/abs/2012.07436). Hence, the results of Toto and the other results that are taken from earlier manuscripts are not comparable.

I would kindly ask the reviewer to perform the experiments under the same condition as the manuscripts where the experimental results are reported from. I will consider raising my score if the experiments are updated or it is clarified that the experiments are indeed performed under the same conditions.

**Questions:**

I have no further questions.

---

### Official Review · Reviewer_wLgd · 2024-11-01

**Soundness:** 3
**Presentation:** 3
**Contribution:** 2
**Rating:** 5
**Confidence:** 4

**Summary:**

This paper introduces Toto, a foundation model designed for time series forecasting, with a specific emphasis on observability metrics. Toto features two core innovations: a proportional factorized attention mechanism, which groups multivariate time series features to reduce computational overhead, and a Student-T mixture model head, which improves the model’s ability to capture complex time series dynamics beyond what traditional Gaussian models offer. Trained on one trillion data points, including a large dataset of proprietary observability metrics, Toto achieves state-of-the-art zero-shot performance in both general benchmarks and observability-specific tasks.

**Strengths:**

**S1:** The writing style is concise, and the methodology is well-organized and clearly described.

**S2:** The paper introduces a new dataset, a large dataset of proprietary observability metrics, which contains statistical characteristics absent from existing datasets.

**S3:** The study employs a dual perspective in the attention mechanism (space-wise and time-wise), and uses this model for pre-training a foundation model to validate its performance.

**Weaknesses:**

**W1:** The paper’s two primary contributions are the novel observability data and the newly designed foundation model. However, neither the data nor the model are publicly available, raising concerns about reproducibility and broader applicability.

**W2:** The technical novelty is limited; for instance, the probabilistic forecasting using the Student-T mixture model (SMM) is an extension, as the Student-T distribution as a prediction head has already been proposed [1].

**W3:** In Table 1, the baseline results are cited directly from Woo et al. (2024) and Das et al. (2024). However, the current information provided does not allow for adequate evaluation.

**W4:** The experiments lack essential details, such as the historical context length in Section 4.1, the sampling strategy and quantity, and the configurations for historical context length and prediction length in Section 4.2.

**W5:** The experimental evaluation is insufficient, omitting several important baselines. Additionally, the proposed method lacks a parameter sensitivity analysis.

**Questions:**

- In Experiment 4.1, several baselines are missing, such as Chronos [2] and Timer [3] in the zero-shot setting, and TimeMixer [4] in the full-shot setting. Similarly, Experiment 4.2 lacks Timer as a baseline.

- What are the historical context window lengths for the baseline models in Experiment 4.1? In Experiment 4.2, what are the historical context windows and prediction lengths for Toto and the other baselines?

- The sampling strategy for Chronos and Moirai models in Experiment 4.2 is unclear. Why is the original sampling strategy from the referenced papers followed? Additionally, is a sampling quantity of 20 fair for Chronos?

- What is the sampling quantity for Toto in Experiment 4.2? Is it also set to 200? How would increasing the sampling quantity affect the baselines' performance?

- In Line 303, it is stated that "for more precise tails, we can choose to spend more computation on sampling"; however, this aspect is not demonstrated in the experiments.


[1] Woo, Gerald, et al. "Unified training of universal time series forecasting transformers." arXiv preprint arXiv:2402.02592 (2024).

[2] Ansari, Abdul Fatir, et al. "Chronos: Learning the language of time series." arXiv preprint arXiv:2403.07815 (2024).

[3] Liu, Yong, et al. "Timer: Transformers for time series analysis at scale." arXiv preprint arXiv:2402.02368 (2024).

[4] Wang, Shiyu, et al. "Timemixer: Decomposable multiscale mixing for time series forecasting." arXiv preprint arXiv:2405.14616 (2024).

---

### Official Review · Reviewer_zDmM · 2024-11-03

**Soundness:** 1
**Presentation:** 2
**Contribution:** 2
**Rating:** 3
**Confidence:** 5

**Summary:**

This paper proposes a novel Transformer-based forecasting model tailored to observability, focusing on metric time series from system monitoring. Key contributions include 1/ a novel transformer block integrating one channel-mixing attention layer followed by multiple blocks of time-wise attention; and 2/ an output head employing a mixture of Student-t distributions. Using a decoder-only architecture, the authors train the model on proprietary observability data, supplemented by open-source and synthetic datasets. The model’s effectiveness is evaluated on both public datasets and a private observability benchmark.

**Strengths:**

* The application to time series forecasting in observability is intriguing and valuable. A step toward unified benchmarking and a useful dataset is commendable.
* The design of proportional factorized space-time attention potentially offers an expressive and efficient backbone for multivariate time series modeling.

**Weaknesses:**

* The work lacks clear motivation; the authors’ objective is difficult to discern.
  * If the aim is a superior observability model, why opt for a foundation model? Any deep forecasting backbone could be trained on similar data for comparison. Instead, only zero-shot models are evaluated on the observability benchmark, where presumably the proposed model was pre-trained.
  * If the goal is to demonstrate generalization of the model trained on observability data to other domains, more comprehensive benchmarks are necessary (e.g., Chronos benchmark; LSF is overly simplistic).
  * If the focus is on the efficiency of the factorized space-time attention mechanism, standard task-specific training regimes would suffice instead of pre-training and zero-shot evaluation.
 * If domain-specific training data is a contribution, details are needed on the pre-training corpus construction, ablation studies, and novelty of data selection.
* The paper’s presentation could be significantly improved in terms of completeness, clarity, and structure.
* The technical contributions of space-time attention and Student-t mixture heads are of limited novelty and provide marginal improvement.
  * Experimental results are insufficient to substantiate the claimed contributions:
  * There is no empirical evidence of the computational efficiency of the proposed attention module.
  * For the observability benchmark, only zero-shot model results are presented. Given the extensive pre-training on observability data, fine-tuned versions of the baseline models would provide a fairer comparison. Additional baselines (e.g., classical models, task-specific deep learning models such as TFT, TiDE) should be included.

**Questions:**

See above.

---

### Note · Authors · 2024-11-19

I have read and agree with the venue's withdrawal policy on behalf of myself and my co-authors.